# Species-specific relationships between net primary productivity and forest age for subtropical China

Peng Li<sup>1</sup>, Rong Shang<sup>1,2\*</sup>, Jing M. Chen<sup>1,3\*</sup>, Huiguang Zhang<sup>4</sup>, Xiaoping Zhang<sup>4</sup>, Guoshuai Zhao<sup>4</sup>, Hong Yan<sup>4</sup>, Jun Xiao<sup>4</sup>, Xudong Lin<sup>1</sup>, Lingyun Fan<sup>1</sup>, Rong Wang<sup>1</sup>, Jianjie Cao<sup>1</sup>, and Hongda Zeng<sup>1</sup>

- <sup>1</sup> Key Laboratory for Humid Subtropical Eco-Geographical Processes of the Ministry of Education, School of Geographical Sciences, Fujian Normal University, Fuzhou, 350117, China
  - <sup>2</sup> Academy of Carbon Neutrality, Fujian Normal University, Fuzhou 350117, China
  - <sup>3</sup> Department of Geography and Planning, University of Toronto, Ontario, ON M5S 3G3, Canada
  - <sup>4</sup> Fujian Forestry Survey and Planning Institute, Fuzhou 350003, China
- 10 Correspondence to: Rong Shang (shangrong@fjnu.edu.cn) and Jing M. Chen (jing.chen@utoronto.ca)

Abstract. The relationship between net primary productivity (NPP) and forest age varies among forest species, yet there were no available NPP-age relationships established for various forest species in subtropical China for use in forest carbon modeling. This study explored the NPP-age relationships for seven forest species in subtropical China using field survey data from the Strategic Priority Project of Carbon Budget (SPPCB), National Forest Inventory (NFI) Type I (NFI-I), and Type II (NFI-II) data. Forest species included Pinus massoniana (P. massoniana), Cunninghamia lanceolata (C. lanceolata), Eucalyptus robusta (Eucalyptus), Other Coniferous Forests (OCF), Softwood Broadleaf (SWB), Hardwood Broadleaf excluding Eucalyptus (HWB), and Mixed Forests (MF). Based on these three datasets, we were able to derive subtropical forest species-specific NPP-age relationships using the Semi-Empirical Model (SEM). Implementation of these species-specific relationships in the process-based Integrated Terrestrial Ecosystem Carbon Cycle (InTEC) model markedly improved above-ground biomass (AGB) simulations for subtropical forests relative to simulations driven by the previously published China-wide NPP-age relationships. The greatest improvements were observed for P. massoniana, OCF, Eucalyptus, and SWB, where root-mean-square errors (RMSE) declined by 19.1–53.3%. These species-specific NPP-age relationships therefore provide a robust, spatially explicit basis for forest carbon modeling and management in subtropical China.

#### 1. Introduction

Forests, recognized as one of Earth's largest carbon sinks, play a crucial role in mitigating climate change and regulating the global carbon cycle (Friedlingstein et al., 2020; Hicke et al., 2007; Yingchun et al., 2012; Eggleston et al., 2006; Pan et al., 2011). Through the process of photosynthesis, forests absorb atmospheric carbon dioxide and convert it into organic carbon, thereby reducing greenhouse gas concentrations in the atmosphere (Chapin et al., 2006; Shang et al., 2023). Specifically, net primary productivity (NPP) serves as a key indicator of forest's carbon sequestration capacity, directly reflecting the biomass accumulation and carbon storage ability of forest ecosystems (Zha et al., 2013; Zhao and Zhou, 2005). NPP exhibits notable

variations with forest age progression (Ben et al., 2004; Wang et al., 2007, 2011). Typically, forest NPP follows a pattern of rapid growth during the early stages, peaking in the middle ages, and gradual decline at old ages (Yu et al., 2017; He et al., 2012). However, the NPP variation pattern with age varies with forest species and climate conditions (Yu et al., 2017; Wang et al., 2018), highlighting the importance of understanding NPP–age relationships across various forest species and climate zones for accurate forest carbon modeling (Yu et al., 2017; Wang et al., 2018; Li et al., 2024b) and effective forest management and ecological restoration strategies (Luyssaert et al., 2008; Li et al., 2024a).

China's subtropical region, characterized by a warm and humid climate, fertile soil, and rich biodiversity, is a vital component of global forest ecosystems (Zhou et al., 2014). This region contributes approximately 67.13% of China's terrestrial ecosystem carbon sequestration (Chen et al., 2019). The subtropical region boasts a diversity of forest species, including evergreen broad-leaved forests, coniferous forests, and mixed forests, with species such as Pinus massoniana (P. massoniana), Cunninghamia lanceolata (C. lanceolata), and Eucalyptus robusta (Eucalyptus) exhibiting unique ecological characteristics that influence the forest NPP–age relationships (Huang et al., 2010). In particular, evergreen broad-leaved and coniferous forests in this region play crucial roles in the forest carbon cycle. Therefore, exploring the NPP–age relationships for diverse forest species in subtropical China is essential for formulating regional forest management strategies.

Currently, two sets of national-scale forest NPP-age curves have been established for China (Li et al., 2024a; Wang et al., 2018). However, these NPP-age curves are constructed based on broad forest cover types such as coniferous forests, broad-leaved forests, or mixed forests, without considering species-specific differences within the same forest type. This limits their application in simulating forest carbon sequestration at the stand-scale and species level. Preliminary research on the NPP-age relationship in subtropical China's forests has been conducted only in Zhejiang Province, distinguishing only between coniferous and broad-leaved forest species (Zheng et al., 2019). Although this study provides a preliminary understanding of the forest carbon cycle in subtropical regions, it fails to account for the diversity of forest species in subtropical areas, especially at the species level, limiting its applicability to national-scale subtropical forest carbon sequestration simulations.

Moreover, constructing species-specific NPP-age curves faces inherent limitations due to insufficient field measurements or survey sample data. For example, field survey samples from the Strategic Priority Project of Carbon Budget (SPPCB) (Fang et al., 2018) and China's National Forest Inventory (NFI) Type I (NFI-I) sample data (Lin et al., 2023) may not ensure an even distribution across different forest age classes. The lack of samples from old-age forest classes hinders the accurate depiction of NPP changes with forest age, leading to biases in carbon sequestration simulations for these classes. The NFI Type II (NFI-II) stand data provides comprehensive coverage of all stands, complementing the insufficient representativeness of sample data, especially for old-age forest samples (Lin et al., 2023). However, NFI-II data typically represent the average conditions of the entire stand or dominant species, which may introduce biases in heterogeneous stands (Lin et al., 2023). Therefore, it is crucial to consider the representativeness of samples and the average values of NFI-II stand data, comprehensively assessing the impact of integrating NFI-II stand data on the construction of NPP-age curves to determine the optimal approach for constructing final curves.

This study aims to explore forest NPP-age relationships for different forest species in subtropical China, with three objectives: (1) to comprehensively assess the impact of integrating NFI-II stand data on the construction of NPP-age curves; (2) to explore the NPP-age relationships of diverse forest species in subtropical China; and (3) to evaluate whether the forest species-specific NPP-age relationships can improve forest aboveground biomass modeling. Here, "forest species" denotes a functional-typological classification that groups individual tree species into ecologically and management-relevant categories rather than to biological species in the strict taxonomic sense. The forest species examined in this study include P. massoniana, C. lanceolata, Eucalyptus, Other Coniferous Forests except for P. massoniana and C. lanceolata (OCF), Softwood Broadleaf (SWB), Hardwood Broadleaf excluding Eucalyptus (HWB), and Mixed Forests (MF). Each species is defined by dominant taxa or shared functional or silvicultural traits, thereby enabling robust parameterisation of NPP-age relationships across heterogeneous subtropical forest stands. The resulting species-specific NPP-age relationships will provide scientific support for estimating forest carbon sequestration and formulating forest management strategies in subtropical China, contributing to enhanced understanding and management of forest carbon dynamics in this region.

# 2. Study Area, Data and Methods

## 2.1. Study Area

Fujian Province was selected as the study area (Fig. 1a) because of its highest forest coverage in China and data availability (Shang et al., 2025). It is located on the southeastern coast, ranging from 23°33′N to 28°20′N in latitude and from 115°50′E to 120°40′E in longitude. The province is predominantly mountainous, with over 80% of its terrain comprising hills and mountains, ranging in elevation from approximately 1500 meters in the northwest to around 500 meters in the southeast. Fujian experiences a subtropical monsoon climate, characterized by mean annual temperatures ranging from 17°C to 21°C and annual precipitation between 1400 mm and 2000 mm. Fig. 1a shows the spatial distribution of the merged forest species. Three key forest species were directly selected for analyzing forest NPP in relation to age: P. massoniana (27.54% of the total studied forest species), C. lanceolata (23.35%), and Eucalyptus (4.14%). P. massoniana and C. lanceolata were chosen for their extensive distribution within Fujian Province (Lin et al., 2023). Eucalyptus, although representing a smaller proportion of the forest, was included due to its artificial continuity, rapid growth, high yield, and economic value (Zhou and Wingfield, 2011). The remaining species were merged into four groups: HWB (26.45%), SWB (2.01%), OCF (2.44%), and MF (14.07%). Bamboo species were not discussed in this study.

Figure 1: The distribution of forest species in Fujian Province (a) and the distribution of NFI-I, NFI-II, and SPPCB field survey samples (b). Different colours indicate different forest species, and the grey colour is for bamboo. P. massoniana: Pinus massoniana, C. lanceolata: Cunninghamia lanceolata, Eucalyptus: Eucalyptus robusta smith, HWB: Hardwood Broadleaf excluding Eucalyptus, SWB: Softwood Broadleaf, OCF: Other Coniferous Forests excluding P. massoniana and C. lanceolata, MF: Mixed Forests.

# 2.2. Data




Forest field data from China's National Forest Inventory (NFI), comprising Type I point data (NFI-I) and Type II polygon data (NFI-II), along with SPPCB field survey samples (Fang et al., 2018), were used for building the forest NPP–age relationships. Fig. 1b shows the spatial distribution of the NFI-I, NFI-II, and SPPCB forest field samples, represented by different colors. The SPPCB field survey samples have previously been effectively used for constructing ten forest NPP–age relationships across China (Li et al., 2024a; Shang et al., 2023) and we only selected the 128 samples located in Fujian for the analysis. It records the sample location, survey time (from 2009 to 2013), forest cover type, age, forest aboveground and underground biomass (Li et al., 2024a). The ground survey size for each SPPCB sample was 1000 m² (600 m² for some plantations), closely approximating a 30-m resolution (Lin et al., 2023). NFI-I samples were obtained from China's 8th (2009-2013) and 9th (2014-2018) National Forest Inventories. Each NFI-I sample records various attributes, including survey time and location, dominant forest species, forest height, diameter at breast height (DBH), forest stock volume, average forest age, and so on. The ground

survey size for each NFI-I sample is typically 667 m<sup>2</sup> (1 mu, a square of 25.82 m × 25.82 m), closely approximating a 30-m resolution. After screening for different forest species, a total of 2,746 samples were retained for each period.



Given the limited availability of NFI-I and SPPCB samples, these data might be insufficient to effectively constrain the NPP-age curve in older forest age ranges. Consequently, we incorporated NFI-II polygons into our analysis. These polygons were rasterized into 30 m spatial resolution pixels using the nearest neighbor resampling method, and all pixels within a forest polygon shared the same attributes (Lin et al., 2023). The NFI-II samples were then created based on the dominance of forest species, requiring a proportion of 100% (with an adjustment to "higher than 80%" for C. equisetifolia and SWB due to their sample sizes) and the availability of relevant attribute records necessary for establishing forest NPP-age relationships (Lin et al., 2023). To ensure sample homogeneity and confirm that each sample is positioned at the center of the forest polygon, all adjacent 11×11 pixels were required to meet both criteria (Lin et al., 2023). Finally, NFI-II samples with old ages combined with the NFI-I and SPPCB samples (Fig. 1b and Fig. 2) were used to construct the forest NPP-age relationships. For each forest species, 80% of samples were randomly selected for building the NPP-age relationships, and the remaining 20% were used to validate the modeled aboveground biomass by the InTEC model with these relationships (see section 2.3.2 for details).

Figure 2: Age distributions of the seven forest species based on the NFI-I, NFI-II, and SPPCB field survey samples in Fujian Province.

P. massoniana: Pinus massoniana, C. lanceolata: Cunninghamia lanceolata, Eucalyptus: Eucalyptus robusta smith, HWB: Hardwood Broadleaf excluding Eucalyptus, SWB: Softwood Broadleaf, OCF: Other Coniferous Forests except for P. massoniana and C. lanceolata, MF: Mixed Forest.

# 125 2.3. Methods


# 2.3.1. Building NPP-age relationships for different forest species

The forest field NPP was calculated from the three types of forest field samples, and it consisted of four components: total biomass increment, mortality, foliage turnovers, and fine root turnovers in the soil (Chen et al., 2002; He et al., 2012; Xia et al., 2019; Li et al., 2024a):

$$NPP = dB_c + M + L_f + L_{fr}$$
 (1)

where  $dB_c$  is the annual increment of total living biomass (including stems, branches, and coarse roots); M is mortality ignored in this study due to a lack of observations at the ground plots and its small proportion to NPP (Li et al., 2024a);  $L_l$  is the turnover of leaves per year; and  $L_{fr}$  is the turnover of fine roots per year in the soil. All three NPP components vary with stand age. Among them, the annual increment of total living biomass is the dominant contributor, whereas foliage and fine-root turnover are also indispensable parts of NPP (Li et al., 2024a; He et al., 2012).

The annual increment of total living biomass was calculated from the annual biomass change (dB) and the ratio of carbon content (Li et al., 2011; White et al., 2000; Wu et al, 2016; Xia et al., 2019):

$$dB_c = dB \times c \tag{2}$$





where dB is the annual biomass change and c is the species-specific carbon content in biomass (see Table 1 for the constant values). Biomass was not directly provided in the NFI-I and NFI-II samples, but it could be calculated from the forest volume (V) using species-specific biomass regression equations. The coefficients for these regression equations are presented in Table 1 (Li et al., 2011; Wu et al., 2016). For the SPPCB samples, which were not resurveyed over time, annual biomass changes were estimated with the space-for-time substitution method (Ma et al., 2017; Liu et al., 2024). To reduce the influence of other factors and ensure that the observed biomass change is primarily attributed to stand age, pairs of samples used to calculate dB were restricted to the same forest species, located within 5 km of each other, and differing by no more than 3 years in stand age.

The turnovers of leaves and fine roots per year in the soil could be calculated as follows (Chen et al., 2002; He et al., 2012; Li et al., 2024a):

$$L_l = \frac{LAI}{SLA} \times t_l \times c \tag{3}$$

$$150 L_{fr} = R_{fr,l} \times L_l (4)$$

where LAI is the annual maximum of leaf area index (LAI) downscaled from the GLOBMAP Version 3 LAI product (see section 2.3.2 for details) (Liu et al., 2012), SLA is the specific leaf area,  $t_l$  is the foliage turnover ratio, c is the species-specific carbon content in biomass (same as that in Equation 2), and  $R_{fr,l}$  represents the ratio of carbon allocated to new fine roots to carbon in new leaves. The detailed values for the coefficients of SLA,  $t_l$ , and  $R_{fr,l}$  for different forest species were provided in Table 2 (Li et al., 2024a; Li et al., 2007; White et al., 2000; Xie et al., 2022; Zhou et al., 2008). For HWB, SWB, OCF, and MF,  $t_l$  was assigned evergreen-species values because deciduous samples constitute only 2.23 % of the total samples. The age-related dynamics in  $L_l$  and  $L_{fr}$  are mainly reflected by the age-related dynamics of the annual maximum LAI (Li et al., 2024a; He et al., 2012).

The semi-empirical mathematical (SEM) function (Chen et al., 2003; He et al., 2012; Li et al., 2024a) was used to build the forest NPP-age relationships for different forest species based on the calculated forest field NPP, as it was demonstrated as the optimal method for building NPP-age curves in China (Li et al., 2024a):




$$NPP(x) = a[1 + (b(x/c)^{d} - 1)/e^{(x/c)}]$$
(5)

where NPP(x) is NPP at the age of x, and a, b, c, and d are the coefficients of the SEM function. The uncertainty analysis of the built NPP-age relationships was conducted using the same method in the research of Li et al., (2024a).

Table 1: The coefficients of the species-specific biomass regression equations and carbon content (Li et al., 2011; Wu et al., 2016). B:
Biomass; V: Volume. P. massoniana: Pinus massoniana, C. lanceolata: Cunninghamia lanceolata, Eucalyptus: Eucalyptus robusta smith,
HWB: Hardwood Broadleaf excluding Eucalyptus, SWB: Softwood Broadleaf, OCF: Other Coniferous Forests except for P. massoniana
and C. lanceolata, MF: Mixed Forests.

| Forest species | Biomass regression equation (t·ha-1) | Carbon content (%) |
|----------------|--------------------------------------|--------------------|
| C. lanceolata  | B = 0.3999V + 22.5410                | 51.27              |
| P. massoniana  | B = 0.52V                            | 52.71              |
| OCF            | B = 0.4631V + 24.2777                | 51.68              |
| Eucalyptus     | B = 0.7893V + 6.9306                 | 47.48              |
| HWB            | B = 0.6255V + 91.0013                | 49.01              |
| SWB            | B = 0.4754V + 30.6034                | 45.02              |
| MF             | B = 0.8019V + 12.2799                | 48.93              |

Table 2: The input coefficients in the calculation of forest field NPP for different forest species. SLA is the specific leaf area;  $t_l$  is the foliage turnover ratio;  $R_{fr,l}$  is the ratio of NPP to fine roots and leaves. P. massoniana: Pinus massoniana, C. lanceolata: Cunninghamia lanceolata, Eucalyptus: Eucalyptus robusta smith, HWB: Other hardwood broadleaf excluding Eucalyptus, SWB: Softwood broadleaf, OCF: Other coniferous mixed forests excluding P. massoniana and C. lanceolata, MF: Mixed forest.

| Forest species | <b>SLA</b> (m <sup>2</sup> kg C <sup>-1</sup> ) | t <sub>l</sub> (year <sup>-1</sup> ) | $R_{fr,l}$ (kg C kg C <sup>-1</sup> ) |
|----------------|-------------------------------------------------|--------------------------------------|---------------------------------------|
| C. lanceolata  | 7.9                                             | 0.22                                 | 1.4                                   |
| P. massoniana  | 6.7                                             | 0.26                                 | 1.4                                   |
| OCF            | 8.2                                             | 0.26                                 | 1.4                                   |
| Eucalyptus     | 26.3                                            | 0.86                                 | 1.2                                   |
| HWB/SWB        | 32                                              | 0.86                                 | 1.2                                   |
| MF             | 21.1                                            | 0.56                                 | 1.3                                   |

#### 2.3.2. Forest carbon modeling using the newly built NPP-age relationships

The NPP-age relationships constructed for different forest species were integrated into the Integrated Terrestrial Ecosystem Carbon Cycle (InTEC) model for forest carbon modeling. To evaluate whether the forest species-specific NPP-age relationships can improve forest carbon modeling, the forest carbon modeling using the newly built NPP-age relationships was compared with that of using the China-wide NPP-age relationships (Shang et al., 2023; Li et al., 2024a). The InTEC model integrates multiple processes, including leaf photosynthesis (using the Farquhar biochemical model), soil carbon and

nitrogen cycling, net nitrogen mineralization, and NPP-age relationships (Chen et al., 2000a, b). This model estimates forest carbon balance by accounting for atmospheric, climatic, and biological changes since the pre-industrial era. The impact of climate change on photosynthesis is modeled through changes in the growing season length and photosynthetic rate, while elevated CO<sub>2</sub> concentrations and leaf nitrogen content positively affect photosynthesis. Model inputs include spatially distributed data on climate, soil texture, nitrogen deposition, and vegetation parameters derived from remote sensing (Table 3). Climate, atmospheric composition, and soil data with resolutions coarser than 30 m were resampled to 30 m using nearest-neighbor resampling. Given the coarse resolution of the climate data, the empirical formulas embedded in the BEPS-TerrainLab model (Xie et al., 2023; Govind et al., 2009) were applied to adjust the resampled climate data using elevation, slope, aspect, and solar position, thereby mitigating the impacts of both resolution and topography. The 30 m NPP generated by the Biosphere-atmosphere Exchange Process Simulator (BEPS) model for 2015, incorporating topographic effects (Cao et al., 2025), served as the reference NPP. The annual maximum LAI, originally from the 500-m GLOBMAP LAI V3 product, was downscaled to 30 m using the Reduced Simple Ratio (RSR) derived from Landsat data—an index used for LAI retrieval (Liu et al., 2012).

$$LAI_{30} = RSR_{30}/RSR_{500} \times LAI_{500} \tag{5}$$

$$RSR = \rho_{NIR}/(\rho_{NIR} + \rho_{SWIR1}) \tag{6}$$

where  $LAI_{30}$  and  $LAI_{500}$  are the annual maximum LAI at 30 m and 500 m resolution, respectively;  $RSR_{30}$  and  $RSR_{500}$  are the corresponding RSR at 30 m and 500 m resolution;  $\rho_{NIR}$  and  $\rho_{SWIR1}$  are Landsat surface reflectance in the near-infrared and short-wave infrared 1 bands.

Forest carbon modeling was conducted from 1986 to 2023 at a 30 m resolution. The period from 1901 to 1985 was used to spin up the soil carbon pools, reducing uncertainties in subsequent simulations. Specifically, the InTEC model assumes that the forest carbon cycle was in equilibrium before the Industrial Revolution, with NPP equaling heterotrophic respiration (Chen et al., 2000a, b). The model iterates using historical climate and atmospheric composition data, allowing the soil carbon pools to gradually adjust to a realistic and stable state, thereby reflecting long-term ecological dynamics prior to the study period (Chen et al., 2000a, b). Initializing the soil carbon pools in this way reduces the model's sensitivity to arbitrary initial conditions, yielding more robust and reliable transient simulation results.

The performance of forest carbon modeling was indirectly validated by comparing the modeled aboveground biomass (AGB) with the calculated AGB from forest field surveys or inventory data, since carbon flux measurements were not available in Fujian province. For each forest species, 20% of samples were randomly selected for validation. Both the SPPCB and NFI-I samples have a survey size closely approximating a 30 m resolution (Lin et al., 2023), while the NFI-II samples, though potentially larger than 30 m, were strictly screened and constrained to be located at the center of homogeneous forest polygons. Given the potential for significant AGB differences across different age groups, a stratified random sampling strategy was


employed to select the validation samples. Specifically, validation samples were randomly selected within each 10-year age group to ensure adequate representation across all age groups. This approach ensured that the validation process was robust and representative of the full range of forest ages, thereby providing a comprehensive assessment of model performance across the entire age spectrum of the forest stands.

Table 3: Input Data of the InTEC Model. LAI: Leaf area index; BEPS: Biosphere-atmosphere Exchange Process Simulator; DEM: Digital elevation model.

| Input data              |                               | Unit                                   | Spatial resolution | Temporal resolution | Data source                                                                                          |
|-------------------------|-------------------------------|----------------------------------------|--------------------|---------------------|------------------------------------------------------------------------------------------------------|
|                         | Precipitation                 | mm                                     |                    |                     |                                                                                                      |
| Climate data            | Temperature                   | $^{\circ}\mathrm{C}$                   | 0.5°               | 1901-2023           | CRU TS 4.08 (Harris et                                                                               |
|                         | Vapor pressure                | hpa                                    |                    |                     | al., 2020)                                                                                           |
|                         | Cloud amount                  | %                                      |                    |                     |                                                                                                      |
| Atmospheric composition | CO <sub>2</sub> concentration | mol mol <sup>-1</sup>                  | Site scale         | 1960-2021           | Mauna Loa (Keeling et al., 1976)                                                                     |
| data                    | Nitrogen deposition           | 10*gN m <sup>-2</sup> yr <sup>-1</sup> | 1.27°×2.5°         | 1997-2013           | (Gao et al., 2020)                                                                                   |
|                         | Forest cover types            | /                                      | 30m                | /                   | NFI-II (Lin et al., 2023)                                                                            |
| Vegetation              | LAI                           | $m^2 m^{-2}$                           | 500m               | 2015                | GLOBMAP LAI V3 (Liu et al., 2012)                                                                    |
| data                    | Forest age                    | year                                   | 30m                | 2015                | NFI-II (Lin et al., 2023)                                                                            |
|                         | Reference NPP                 | 10 gC m <sup>-2</sup> yr <sup>-1</sup> | 30m                | 2015                | BEPS (Cao et al., 2025)                                                                              |
|                         | NPP-age curves                | /                                      | /                  | /                   | This study                                                                                           |
|                         | Sand content                  | %                                      | 0.0083°            | /                   | HDSW World Soil                                                                                      |
| Soil data               | Clay content                  | %                                      | $0.0083^{\circ}$   | /                   | Database (FAO and                                                                                    |
|                         | Soil depth                    | 100 m                                  | $0.0083^{\circ}$   | /                   | IIASA, 2023)                                                                                         |
|                         | Latitude/longitude            | degree                                 | 30m                | /                   | /                                                                                                    |
| Topographic<br>data     | DEM                           | m                                      | 30m                | /                   | ASTER GDEM<br>(NASA/METI/AIST/Jap<br>an Space Systems and<br>U.S./Japan ASTER<br>Science Team, 2018) |
|                         | Slope and aspect              | /                                      | 30m                | /                   | Science Team, 2016)                                                                                  |
|                         | • •                           | /                                      | 30111              | /                   |                                                                                                      |
|                         | Topographic wetness index     | /                                      | 30m                | /                   | Calculated from DEM                                                                                  |
|                         | Water table depth             | m                                      | 30m                | /                   |                                                                                                      |

# 2.3.3. Comparison between the species-specific and China-wide NPP-age relationships

The NPP-age relationships for seven forest species (referred to as species-specific curves) in Fujian province were compared with the built NPP-age relationships for entire China (shortened to as China-wide curves) (Li et al., 2024a). Previously, ten China-wide NPP-age curves were built by separating the southern and northern regions and five forest cover types (Li et al.,

2024a): evergreen broad-leaved forests (EBF), evergreen needle-leaved forests (ENF), deciduous broad-leaved forests (DBF), deciduous needle-leaved forests (DNF), and mixed forests (MF). Only the southern-region ENF, EBF and MF curves were relevant to Fujian province, so the species-specific curves for C. lanceolata, P. massoniana and OCF were compared against the southern ENF curve, those for Eucalyptus, HWB and SWB against the southern EBF curve, and the MF curve against the southern MF curve. The intrinsic features of the species-specific and China-wide NPP-age curves and their performances within the InTEC carbon modeling were systematically compared.

#### 3. Results







### 3.1. Comparisons of forest NPP-age relationships constructed with and without NFI-II samples

The NPP-age relationships constructed with and without NFI-II samples using the SEM function were compared in Fig. 3. This comparison was motivated by two considerations: (i) NFI-II and SPPCB samples alone may not provide sufficient data to reliably constrain NPP-age curves for old forests and (ii) NFI-II samples, derived from NFI-II polygons, may introduce inherent uncertainties. Their curve-fitting performances were quantitatively assessed using R<sup>2</sup> and RMSE, as shown in Fig. 4. The age of the forest at its maximum NPP (referred to as the peak NPP age), a critical indicator of the NPP-age relationship, remained consistent for Eucalyptus, P. massoniana, C. lanceolata, OCF, and MF, regardless of whether NFI-II samples were

remained consistent for Eucalyptus, P. massoniana, C. lanceolata, OCF, and MF, regardless of whether NFI-II samples were included, while only a one-year difference was observed for HWB and SWB. Notably, without using NFI-II samples, NPP values for C. lanceolata and SWB dropped close to zero in ages older than 150 years, suggesting a transition from forest carbon sinks to carbon sources in old ages. This finding contrasts with previous studies, which suggest that older forests continue to act as carbon sinks (Gundersen et al., 2021; Luyssaert et al., 2008). For Eucalyptus, the NPP reduction exceeded 70% without NFI-II samples, diverging significantly from previous studies that generally report reductions by about one-third (Luyssaert et al., 2008; Wang et al., 2011) or half (Ryan et al., 2004; Mund et al., 2002) of peak NPP. These discrepancies highlight the importance of including NFI-II samples for accurately modeling the NPP–age relationships for Eucalyptus, C. lanceolata, and SWB.

For P. massoniana, HWB, OCF, and MF, the inclusion of NFI-II samples had minimal effects on the overall pattern of the NPP-age curves. When NFI-II samples were included, R<sup>2</sup> slightly decreased by less than 0.025 for P. massoniana, HWB, and OCF, but slightly increased by 0.003 for MF. Similarly, RMSE values showed a minor increase (under 2 gC m<sup>-2</sup> year<sup>-1</sup>) for P. massoniana and OCF, while a slight decrease (under 7 gC m<sup>-2</sup> year<sup>-1</sup>) for HWB and MF. Therefore, we ultimately opted to consistently use NFI-II samples in constructing the NPP-age curves, as incorporating NFI-II samples can extend the age range over which the curves are constrained, thus enhancing the data coverage and consistency.

Figure 3: NPP-age curves fitted by the SEM function for different forest species with and without using NFI-II samples. The green and red lines depict the forest NPP-age curves with and without using NFI-II samples, respectively. Solid lines indicate the forest age ranges where field data are available, while dashed lines represent extrapolated curves beyond the field sample age range. The red and blue circles, with associated grey error bars, represent the average NPP values and their one standard deviation. The green and red lines depict the built forest NPP-age curves with and without using NFI-II samples. Solid lines indicate the forest age ranges where field data are available, while dashed lines represent extrapolated curves beyond the maximum age of the field samples. P. massoniana: Pinus massoniana, C. lanceolata: Cunninghamia lanceolata, Eucalyptus: Eucalyptus robusta smith, HWB: Hardwood Broadleaf excluding Eucalyptus, SWB: Softwood Broadleaf, OCF: Other Coniferous Forests except for P. massoniana and C. lanceolata, MF: Mixed Forest.

Figure 4: R<sup>2</sup> and RMSE of the built NPP-age curves for different forest species with and without NFI-II samples. P. massoniana: Pinus massoniana, C. lanceolata: Cunninghamia lanceolata, Eucalyptus: Eucalyptus robusta smith, HWB: Hardwood Broadleaf excluding Eucalyptus, SWB: Softwood Broadleaf, OCF: Other Coniferous Forests except for P. massoniana and C. lanceolata, MF: Mixed Forest.

#### 3.2. Characterization of forest NPP-age curves among different forest species





The final species-specific forest NPP-age curves were selected from the built curves using all field NPP samples (green lines in Fig. 3), and their coefficients were provided in Table 4. To facilitate a comparative characterization of forest NPP-age relationships among different forest species, these curves were normalized and jointly displayed in Fig. 5. Solid lines indicate the age range supported by field data (the triangle in each line indicates the maximum age), while dashed lines indicate predicted values beyond this range using the SEM function. The NPP-age patterns were generally consistent across all species, with NPP increasing during young stages, peaking in a middle age, and then declining and stabilizing in old ages (Li et al., 2024a; He et al., 2012; Yu et al., 2017; Zheng et al., 2019; Wang et al., 2018, 2011). But there were also variations in the timing of peak NPP, as well as differences in the rate of decline in older age stages. Specifically, the peak NPP ages for Eucalyptus, P. massoniana, C. lanceolata, SWB, HWB, OCF, and MF were identified as 9, 32, 25, 22, 37, 24, and 30 years, respectively. The ratios of stabilized NPP in old ages to the maximum NPP (stabilized-to-peak NPP ratios) were 59.1%, 57.3%, 65.5%, 56.8%, 59.5%, 56.0%, and 57.9%, respectively. These values align with previous studies, which typically report reductions by approximately one-third (Luyssaert et al., 2008; Wang et al., 2011) or half (Ryan et al., 2004; Mund et al., 2002) from the peak NPP.

Broadleaf species such as HWB and SWB demonstrated later peak NPP ages and lower stabilized-to-peak NPP ratios compared to conifer species like C. lanceolata, P. massoniana, and OCF. The higher wood density and longer lifespans of broadleaf species allow them to sustain productivity and carbon absorption over an extended period (Xu et al., 2024), while conifer species, despite their rapid early growth and carbon fixation, show earlier and steeper declines, reflecting differences

in their ecological and physiological strategies (Bigler and Veblen, 2009). HWB exhibited a later peak NPP age and lower stabilized-to-peak NPP ratio compared to SWB. This can be explained by the fact that hardwood species maintain stronger carbon absorption during later growth stages due to their higher wood density and longer lifespans (Luyssaert et al., 2008; Mun et al., 2020). In contrast, softwood species excel in rapid carbon sequestration during early stages but experience earlier and more significant productivity declines (Stephenson et al., 2014). Compared to C. lanceolate, P. massoniana shows a later peak NPP age and higher stabilized-to-peak NPP ratio, as P. massoniana supports prolonged carbon sequestration (Justine et al., 2017; Bai and Ding, 2024), while C. lanceolata prioritizes rapid early growth (Zhou et al., 2016).




The NPP-age curve for Eucalyptus forests exhibits a relatively low peak NPP age of 9 years compared to other forest species. This early peak age underscores the species' ability to achieve significant productivity at a young age, making it well-suited for fast-growing timber plantations (Zhang et al., 2023; Qin and Shangguan, 2019). While this early productivity surge is advantageous for short-rotation forestry, it often leads to a shortened early life cycle, causing a noticeable reduction in productivity soon after reaching peak levels (Zhang et al., 2023; Zhou and Wingfield, 2011).

Table 4: Coefficients of species-specific forest NPP-age curves built by the SEM function in Fujian Province. a-d: the coefficients. P. massoniana: Pinus massoniana, C. lanceolata: Cunninghamia lanceolata, Eucalyptus: Eucalyptus robusta smith, HWB: Hardwood Broadleaf excluding Eucalyptus, SWB: Softwood Broadleaf, OCF: Other Coniferous Forests except for P. massoniana and C. lanceolata, MF: Mixed Forest.

| Encort and aire | Parameters |       |       |       |
|-----------------|------------|-------|-------|-------|
| Forest species  | а          | b     | С     | d     |
| C. lanceolata   | 470.8      | 1.894 | 13.78 | 1.353 |
| P. massoniana   | 415.3      | 0.132 | 7.61  | 4.136 |
| OCF             | 422        | 1.325 | 9.697 | 2.267 |
| Eucalyptus      | 484        | 1.225 | 3.664 | 2.226 |
| HWB             | 507.2      | 1.132 | 14.38 | 2.316 |
| SWB             | 468        | 0.532 | 6.847 | 3.104 |
| MF              | 444.2      | 1.246 | 12.11 | 2.26  |

Figure 5: The normalized NPP-age curves built from the SEM function. The solid lines are for the age period with field data (the triangle in each line indicates the largest age with the field data), and the dashed lines are for the age period without field data. P. massoniana: Pinus massoniana, C. lanceolata: Cunninghamia lanceolata, Eucalyptus: Eucalyptus robusta smith, HWB: Hardwood Broadleaf excluding Eucalyptus. SWB: Softwood Broadleaf. OCF: Other Coniferous Forests except for P. massoniana and C. lanceolata, MF: Mixed Forest.

### 3.3. Comparison to the forest NPP-age curves built previously





The normalized seven species-specific NPP-age curves were compared with three previously built China-wide curves (Fig. 6). The species-specific curves for C. lanceolata, P. massoniana and OCF were compared against the southern ENF curve, those for Eucalyptus, HWB and SWB against the southern EBF curve, and the MF curve against the southern MF curve. In general, the species-specific NPP-age curves constructed exhibit earlier peak ages and faster decline in old ages, particularly for Eucalyptus, C. lanceolate, OCF, and SWB.

The peak NPP age for Eucalyptus is 9 years, much smaller than that from the China-wide NPP-age curve. But it aligns with the reported rapid growth and significant productivity of Eucalyptus at young ages (Zhang et al., 2023; Qin and Shangguan, 2019). The peak NPP age for P. massoniana in Fujian Province is 32 years, which agrees well with the 34 years of ENF in southern China (Li et al., 2024a). While C. lanceolata peaks at 25 years, earlier than ENF in southern China, but close to the 23 years found for coniferous forests in Zhejiang Province (Zheng et al., 2019). Other coniferous forests except for P. massoniana and C. lanceolate have peak NPP ages of 24 years, similar to coniferous forests in Zhejiang (Zheng et al., 2019). These relatively early peak ages indicate their efficient photosynthesis and resource utilization during the early growth stages (Lu et al., 2015; Huang et al., 2007).

The peak NPP age of HWB is 37 years, similar to that of EBF in southern and eastern China, which peaks at 30 to 40 years (Li et al., 2024a; Wang et al., 2011). In contrast, the NPP of SWB peaks at 22 years, which is relatively early compared to

HWB. Soft broadleaf species generally prioritize rapid early growth in response to favorable environmental conditions (Fujita et al., 2012). Among broadleaf species, Eucalyptus exhibits the earliest peak NPP of 9 years, making it highly suitable for short-rotation forestry, but sharp post-peak declines limit its long-term carbon storage potential. The peak NPP age of MF in Fujian is 30 years, which is similar to that of mixed forests in the south and southwest of China (Li et al., 2024a) and close to the 32 years of peak NPP age for MBF in central China (Wang et al., 2011). Mixed forests combine the fast growth of broadleaf species with the longevity of conifers, achieving a balance in productivity across growth stages. Their diverse composition enhances resource utilization efficiency and reduces competition, allowing for sustained and stable carbon absorption (Xu et al., 2024).

Figure 6: Comparison between the species-specific and China-wide normalized forest NPP-age curves. P. massoniana: Pinus massoniana, C. lanceolata: Cunninghamia lanceolata, Eucalyptus: Eucalyptus robusta smith, HWB: Hardwood Broadleaf excluding Eucalyptus, SWB: Softwood Broadleaf, OCF: Other Coniferous Forests except for P. massoniana and C. lanceolata, MF: Mixed Forest.

# 3.4. Forest biomass modeling using the species-specific NPP-age curves





The built species-specific NPP-age curves were incorporated into the InTEC model for forest biomass modeling. But due to the lack of field soil carbon data for validation, we primarily focused on validating the modeled forest AGB. We compared the simulated AGB obtained by using the newly constructed species-specific NPP-age curves with that obtained by using the previously built nationwide NPP-age curves (Fig.7). Accuracy was evaluated with R<sup>2</sup> and RMSE against the calculated field AGB from a randomly withheld 20 % of the forest field samples, and higher R<sup>2</sup> and lower RMSE indicate better performance. Overall, the species-specific NPP-age curves significantly outperformed the nationwide curves in simulating AGB accuracy.

For coniferous forests, the nationwide NPP-age curve tended to overestimate AGB for ages ranging between 40 and 120 years. In contrast, the species-specific curves declined more rapidly after the peak NPP year. This might be closely related to the mechanism through which the subtropical warm and humid environment accelerates plant physiological aging (Chen et al., 2024). When using species-specific curves, the accuracy of simulating AGB for C. lanceolata was slightly higher, while for both C. lanceolata and OCF, the accuracy was significantly improved, with an average reduction in RMSE ranging from 9.4 to 14.4 Mg/ha. For Eucalyptus and SWB in broadleaf forests, the nationwide curve overestimated AGB for trees older than their peak NPP age but underestimated it for Eucalyptus younger than 20 years. The accuracy of simulating AGB for HWB based on species-specific curves was slightly enhanced, but for Eucalyptus and SWB, it was significantly improved, with an average increase in R<sup>2</sup> greater than 0.3 and a decrease in RMSE exceeding 18.5 Mg/ha. Similarly, the accuracy for MF was also enhanced, with an average RMSE reduction of 7.56 Mg/ha. Overall, the larger the differences between species-specific and nationwide NPP-age relationships (Fig. 6), the larger improvements are found in simulated AGB values (Fig. 7).

These results demonstrate that the newly developed species-specific NPP-age curves significantly enhance the accuracy of AGB simulations in the InTEC model for subtropical forests, particularly for early-maturing species such as Eucalyptus, by capturing regional-specific growth strategies. Notably, the improvement in simulation accuracy varied across different age classes, highlighting the importance of considering age dynamics in forest carbon sink modeling and predictions.

Figure 7: Validation and comparison of the simulated aboveground biomass by using the species-specific and China-wide forest NPP-age curves. P. massoniana: Pinus massoniana, C. lanceolata: Cunninghamia lanceolata, Eucalyptus: Eucalyptus robusta smith, HWB: Hardwood Broadleaf excluding Eucalyptus, SWB: Softwood Broadleaf, OCF: Other Coniferous Forests except for P. massoniana and C. lanceolata, MF: Mixed Forest.

#### 4. Discussions



This study established NPP-age relationships for seven forest species and species groups in Fujian Province based on field survey data from NFI-I, NFI-II, and SPPCB using the SEM model. It also evaluated whether the species-specific NPP-age

relationships can improve forest biomass modelling using the InTEC model. Since NFI-I and SPPCB samples alone do not provide adequate data for old forests, and NFI-II samples introduces inherent uncertainties, we compared NPP-age relationships constructed with and without NFI-II samples. Results showed that incorporating NFI-II samples was crucial for accurately modeling NPP-age relationships for Eucalyptus, C. lanceolata, and SWB, but had minimal impacts on P. massoniana, HWB, OCF, and MF. Nevertheless, NFI-II helped extend the data range to older ages. The constructed species-specific NPP-age relationships all three available datasets were shown to improve modelled biomass in subtropical China, highlighting the importance of species-specific parameterization in forest biomass modeling. The resulting NPP-age curves will provide scientific support for accurate estimation of forest carbon sequestration and the formulation of forest management strategies in subtropical China (Li et al., 2024c), contributing to enhanced understanding and management of forest carbon dynamics in this region with the largest sinks in China.







There were several limitations. Firstly, inherent inconsistencies may arise among the three field data sets, particularly notable discrepancies between the NFI-II stand data and the SPPCB and NFI-I sample data. The NFI-I and SPPCB field samples may lack sufficient representation within the old age classes of forests (Fig. 2), potentially leading to unconstrained NPP-age curves for certain forest species in old ages, which may exhibit an unreasonably declining trend and a transition from forest carbon sinks to carbon sources, i.e. NPP declines to values close to zero (Fig. 3). To strengthen the constraint on the curves for the old age classes, this study incorporated NFI-II stand data by converting stand attributes into point samples. However, for stands characterized by high heterogeneity, deviations may still occur despite efforts to mitigate this effect through screening based on the dominance of forest species (Lin et al., 2023). To visually indicate the data constraint on the constructed NPP-age curves, solid lines were used to denote the curve portions supported by field data, while dashed lines were employed for the curve portions lacking field data. In future studies, collecting more field data on old forests will facilitate determining the shape of the forest NPP-age curves at older ages.

Second, this study did not account for the difference of planted forests and natural forests on the NPP-age relationships, nor the impact of forest managements such as selective logging and shelterwood cutting. Eucalyptus in plantations often grows rapidly due to intensive management, but this can lead to ecosystem degradation, such as soil erosion and reduced biodiversity. In contrast, Eucalyptus in natural forests grows more slowly but supports a more stable ecosystem (Ying et al., 2010). P. massoniana and C. lanceolata in natural forests exhibit higher ecosystem complexity and biodiversity, which results in slower growth rates but longer growth cycles with higher NPP (Liu et al., 2014). The timing of selective cutting and shelterwood cutting also significantly affects forest growth. Properly timed logging practices can promote tree health, growth, and resource renewal, while mistimed logging can negatively impact growth rates and wood quality (Wu et al., 2018). Due to the lack of data, this study did not distinguish between planted and natural forests and did not consider the impact of forest management. Future research may be directed towards acquiring comprehensive data to better understand the growth differences between planted forests and natural forests and the influence of forest managements on forest NPP-age relationships.

Third, the input coefficients for specific leaf area, foliage turnover ratio, and the ratios of the turnovers of fine roots and leaves to NPP used in calculating forest field NPP for diverse forest species may introduce uncertainties into the forest NPP-

age relationships. Currently, these coefficients are primarily sourced from literature (Li et al., 2024a; Li et al., 2007; White et al., 2000; Xie et al., 2022; Zhou et al., 2008), with data originating from subtropical provinces in China such as Guangxi (Xie et al., 2022), Jiangxi (Li et al., 2007), and Guiyang (Zhou et al., 2011), as well as from other regions (White et al., 2000). Data from these regions may differ from those in subtropical China, potentially leading to biases in the calculation of forest field NPP and final built NPP—age curves. Moreover, as deciduous samples constitute only 2.23 % of the total samples, HWB, SWB, OCF and MF were assigned evergreen foliage turnover coefficients. Therefore, future studies should prioritize local field measurements of these key coefficients, particularly for deciduous species, to refine the NPP—age relationships and to quantify the age-dependent carbon sequestration capacity of each species more accurately.

Fourth, there were also other factors that could influence the forest NPP-age relationships, such as the site conditions and soil fertility (Li et al., 2024a). Under favorable site conditions, forests typically exhibit faster NPP growth during their early stages, attain higher peak NPP values, and undergo steeper declines in NPP as they age (Wang et al., 2018; Yu et al., 2017). Conversely, forests with poor soil fertility tend to exhibit slower NPP growth in their early stages, achieve lower peak NPP values, and undergo less dramatic declines in NPP as they mature. Notably, the rapid replacement of natural broadleaf forests with plantations dominated by species such as P. massoniana and C. lanceolata in subtropical regions has significantly reduced soil fertility (Ming et al., 2019; Ni et al., 2021; Li et al., 2023). Therefore, in future research endeavors, it is imperative to consider site conditions and soil fertility to improve the construction of forest NPP-age curves.

Last, the varying spatial resolutions of model inputs may affect the accuracy of model simulations. Downscaling LAI from 500 m to 30 m resolution using the RSR derived from Landsat data helps mitigate some scale-related impacts. However, in complex mountainous terrain, retrieving 30 m LAI may require consideration of additional factors, such as topography. Future research could focus on directly retrieving 30 m LAI based on Landsat data and Global Ecosystem Dynamics Investigation (GEDI) lidar data (Liang et al., 2025), thereby improving model accuracy. Besides, the empirical formulas embedded in the BEPS-TerrainLab V2.0 model (Xie et al., 2023; Govind et al., 2009) were also used to reduce the impacts of coarse resolution climate data. As higher-resolution remote sensing products and more ground climate data become available, it will be possible to integrate higher-resolution climate data to further enhance the performance and reliability of the InTEC forest carbon modeling.

#### 5. Conclusions

This study investigated the NPP-age relationships for seven forest species and species groups in subtropical China, leveraging the extensive datasets from the SPPCB, NFI-I, and NFI-II forest field surveys along with the SEM function. Forest species examined encompassed P. massoniana, C. lanceolata, Eucalyptus, OCF, SWB, HWB, and MF. Given that the NFI-I and SPPCB samples alone might not adequately represent old forests, while the NFI-II samples offer comprehensive coverage across all stands but could potentially introduce inherent uncertainties, we conducted a comparative analysis of the NPP-age curves with and without the inclusion of NFI-II samples. Results showed that incorporating NFI-II samples was crucial for

accurately modeling NPP-age relationships for Eucalyptus, C. lanceolata, and SWB, but had minimal impacts on P. massoniana, HWB, OCF, and MF. Therefore, we incorporated NFI-II samples in constructing the species-specific NPP-age curves to enhance the data coverage and consistency. Significant differences are found between the species-specific and nation-wide NPP-age relationships in both NPP peak age and the ratio of stabilized NPP at old ages to the peak NPP, suggesting dependence of the relationships on forest species and climate.

The built species-specific NPP-age curves were subsequently incorporated into the InTEC model for forest biomass modeling, and results demonstrate that the newly established species-specific curves significantly improved the accuracy of AGB simulations in the InTEC model for subtropical forests, particularly for early-maturing species such as Eucalyptus. Notably, the enhancement in simulation accuracy varied across different age classes, underscoring the significance of considering age dynamics in forest carbon sink modeling and predictions. These species-specific NPP-age curves will serve as a fundamental basis for reliable forest carbon modeling and effective forest management in subtropical China.

# Code availability

The codes for building the forest NPP-age relationships are available upon request from the corresponding authors.

# 440 Data availability

430

435

The coefficients of the built forest NPP-age relationships are available in Table 4.

#### Acknowledgments

Thanks to the two reviewers for their valuable comments and suggestions, as well as to the editors for their assistance with the manuscript.

#### 445 Author contributions

Conceptualization, R.S.; Methodology, P.L., R.S. and J.M.C.; Validation, P.L.; Formal analysis, P.L., X.L and L.F.; Writing—original draft, R.S.; Writing—review & editing, R.S. and J.M.C.; Funding acquisition, R.S.; Data curation, H. Z. (Huiguang Zhang), X. Z., G. Z., H. Y., J. X., W. R., J. C., and H. Z. (Hongda Zeng).

# **Funding sources**

This research was supported by the National Natural Science Foundation of China (42471356, U23A2002, and 42101367), the Fujian Forestry Science and Technology Key Project (2022FKJ03), and the Natural Science Foundation of Fujian Province (2025J09033 and 2024J01469).

#### **Competing interests**

The authors declare that they have no conflict of interest.

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
