# Peer review of "Species-specific relationships between net primary productivity and forest age for subtropical China"

_EGUsphere, 2025_

## Author Response (AR1)

**Summary of revisions:**

We have made major revisions in response to the editor and the two reviewers. Specifically, we: (1) supplied a detialed description about field survey size and time of the SPPCB, NFI-I and NFI-II samples; (2) elaborated on the calculation of field-based NPP; (3) clarified all InTEC model inputs for AGB simulations and documented the harmonization of their spatial resolutions; (4) described the model spin-up procedure; (5) presented the method about the selection of validating field survey samples and the validation of the simulated AGB; (6) added the method about the comparison between the species-specific and China-wide NPP-age relationships; (7) incorporated a discussion on the varying spatial resolutions of model inputs; (8) added the definition of "forest species". Throughout the manuscript, we have also refined wording and added explanatory details to enhance clarity. Moreover, we have followed the reviewers' suggestions and provided feedback to the reviewers on a point-by-point basis (see responses below). All revisions are highlighted in *red* color in the manuscript.

**Response to Editor's comments:**

(*Italic* indicates the manuscript text, *red* indicates revisions)

**Comments E1:**

Please follow the reviewers' suggestions carefully. Especially, please clarify the difference between the used data sets and the used models/equations.

**Response:**

Thanks for your helpful comments, and they were clarifed.

**"Abstract.**

[revised manuscript text omitted]

**Comments E3:**

I suggest also to streamline the usage of the following terms in the manuscript and to define them clearly: 'forest type', 'forest species' and 'tree species'.

**Response:**

Thanks for your valuable comments. We have now consistently applied "forest species", and its definition was provided in the revised Introduction section.

**1. Introduction**

. . . . . .

This study aims to explore forest NPP-age relationships for different forest species in subtropical China, with three objectives: (1) to comprehensively assess the impact of integrating NFI-II stand data on the construction of NPP-age curves; (2) to explore the NPP-age relationships of diverse forest species in subtropical China; and (3) to evaluate whether the forest species-specific NPP-age relationships can improve forest aboveground biomass modeling. Here, "forest species" denotes a functional-typological classification that groups individual tree species into ecologically and management-relevant categories rather than to biological species in the strict taxonomic sense. The forest species examined in this study include P. massoniana, C. lanceolata, Eucalyptus, Other Coniferous Forests except for P. massoniana and C. lanceolata (OCF), Softwood Broadleaf (SWB), Hardwood Broadleaf excluding Eucalyptus (HWB), and Mixed Forests (MF). Each species is defined by dominant taxa or shared functional or silvicultural traits, thereby enabling robust parameterisation of NPP-age relationships across heterogeneous subtropical forest stands. The resulting species-specific NPP-age relationships will provide scientific support for estimating forest carbon sequestration and formulating forest management strategies in subtropical China, contributing to enhanced understanding and management of forest carbon dynamics in this region."

**Response to Reviewer #1's comments**

(*Italic* indicates the manuscript text, *red* indicates revisions)

**Comments 1.1:**

Net primary productivity (NPP) of forests changes with the age. The relationship between NPP and age is crucial for quantifying the carbon sink of forests. The study investigates the species-specific relationships between NPP and forest age over subtropical China on the basis of different sources of field data. Overall, this manuscript is well-written. The topic is interesting. After some

modifications, this manuscript is publishable.

**Response:**

Thanks for your positive feedback.

**Comments 1.2:**

In the calculation of dB, biomass in two different years is required. For the SPPCB dataset, are there biomass values in two different years available?

**Response:**

Thanks for your valuable comments. No, the survey date of the SPPCB samples covers from 2009 to 2019, but they were not resurveyed over time. For the SPPCB samples, we adopted a space-for-time substitution method and pairs of samples used to calculate the annual biomass change were restricted to the same forest species, located within 5 km of each other, and differing by no more than 3 years in stand age. Sections 2.2 and 2.3 were revised accordingly.

**"2.2. Data**

....

The SPPCB field survey samples have previously been effectively used for constructing ten forest NPP-age relationships across China (Li et al., 2024a; Shang et al., 2023) and we only selected the 128 samples located in Fujian for the analysis. It records the sample location, survey time (from 2009 to 2013), forest cover type, age, forest aboveground and underground biomass (Li et al., 2024a). The ground survey size for each SPPCB sample was 1000 m² (600 m² for some plantations), closely approximating a 30-m resolution (Lin et al., 2023).

...."

**"2.3.1. Building NPP-age relationships for different forest species**

. . . . . .

where dB is the annual biomass change and c is the species-specific carbon content in biomass (see Table 1 for the constant values). Biomass was not directly provided in the NFI-I and NFI-II samples, but it could be calculated from the forest volume (V) using species-specific biomass regression equations. The coefficients for these regression equations are presented in Table 1 (Li et al., 2011; Wu et al., 2016). For the SPPCB samples, which were not resurveyed over time, annual biomass changes were estimated with the space-for-time substitution method (Ma et al., 2017; Liu et al., 2024). To reduce the influence of other factors and ensure that the observed biomass change is primarily attributed to stand age, pairs of samples used to calculate dB were restricted to the same forest species, located within 5 km of each other, and differing by no more than 3 years in stand age.

....."

**Comments 1.3:**

Are the values of variable C in equations (2) and (3) the same? Lf is the turnover of leaves per year. Equation (1) assumes that the NPP allocated into leaves is equal to the turnover of leaves for evergreen forests. It means that the foliage carbon does not change annually. This is true for mature evergreen forests. For young evergreen forests, this assumption is questionable to some extent. The foliage carbon of young evergreen forests increases year by year. The NPP allocated into leaves is larger than the turnover loss.

**Response:**

Thanks for your valuable comments.

Yes, the values of variable C in equations (2) and (3) are the same, representing the carbon content within biomass. For each forest species, its value was treated as a constant and is listed in Table 1 of the manuscript.

For Equation (1), the three terms on its right side—change in aboveground biomass (dB), leaf turnover  $(L_l)$ , and fine root turnover  $(L_{fr})$  are all dynamic and change with age. The age-related dynamics in  $L_l$  and  $L_{fr}$  are mainly reflected by the age-related dynamics of the annual maximum LAI.

The manuscript was revised accordingly.

**"2.3.1. Building NPP-age relationships for different forest species**

The forest field NPP was calculated from the three types of forest field samples, and it consisted of four components: total biomass increment, mortality, foliage turnovers, and fine root turnovers in the soil (Chen et al., 2002; He et al., 2012; Xia et al., 2019; Li et al., 2024a):

$$NPP = dB_c + M + L_f + L_{fr} (1)$$

where  $dB_c$  is the annual increment of total living biomass (including stems, branches, and coarse roots); M is mortality ignored in this study due to a lack of observations at the ground plots and its small proportion to NPP (Li et al., 2024a);  $L_l$  is the turnover of leaves per year; and  $L_{fr}$  is the turnover of fine roots per year in the soil. All three NPP components vary with stand age. Among them, the annual increment of total living biomass is the dominant contributor, whereas foliage and fine-root turnover are also indispensable parts of NPP (Li et al., 2024a; He et al., 2012).

The annual increment of total living biomass was calculated from the annual biomass change (dB) and the ratio of carbon content (Li et al., 2011; White et al., 2000; Wu et al., 2016; Xia et al., 2019):

$$dB_c = dB \times c \tag{2}$$

where dB is the annual biomass change and c is the species-specific carbon content in biomass (see Table 1 for the constant values). Biomass was not directly provided in the NFI-I and NFI-II samples, but it could be calculated from the forest volume (V) using species-specific biomass regression equations. The coefficients for these regression equations are presented in Table 1 (Li et al., 2011; Wu et al., 2016). For the SPPCB samples, which were not resurveyed over time, annual biomass changes were estimated with the space-for-time substitution method (Walker et al., 2010; Blois et al., 2013). To reduce the influence of other factors and ensure that the observed biomass change is primarily attributed to stand age, pairs of samples used to calculate dB were restricted to the same forest species, located within 5 km of each other, and differing by no more than 3 years in stand age.

The turnovers of leaves and fine roots per year in the soil could be calculated as follows (Chen et al., 2002; He et al., 2012; Li et al., 2024a):

$$L_l = \frac{LAI}{SLA} \times t_l \times c \tag{3}$$

$$L_{fr} = R_{fr,l} \times L_l \tag{4}$$

where LAI is the annual maximum of leaf area index (LAI) downscaled from the GLOBMAP Version 3 LAI product (see section 2.3.2 for details) (Liu et al., 2012), SLA is the specific leaf area,  $t_l$  is the foliage turnover ratio, c is the species-specific carbon content in biomass (same as that in Equation 2), and  $R_{fr,l}$  represents the ratio of carbon allocated to new fine roots to carbon in new leaves. The detailed values for the coefficients of SLA,  $t_l$ , and  $R_{fr,l}$  for different forest species were provided in Table 2 (Li et al., 2024a; Li et al., 2007; White et al., 2000; Xie et al., 2022; Zhou et al., 2008). For HWB, SWB, OCF, and MF,  $t_l$  was assigned evergreen-species values because deciduous samples constitute only 2.23 % of the total samples. The age-related dynamics in  $L_l$  and  $L_{fr}$  are mainly reflected by the age-related dynamics of the annual maximum LAI (Li et al., 2024a; He et al., 2012)."

**Comments 1.4:**

Leaf area index changes seasonally. Is the annual maximum of LAI used in Equation (3)? Please clarify.

**Response:**

Thanks for your valuable comments. Yes, it is the annual maximum of LAI, and it was revised.

**"2.3.1. Building NPP-age relationships for different forest species**

....

where LAI is the annual maximum of leaf area index (LAI) downscaled from the GLOBMAP Version 3 LAI product (see section 2.3.2 for details) (Liu et al., 2012), ....."

**Comments 1.5:**

As shown in Table 2, the foliage turnover ratio is smaller than 1.0 for all species. Are these species all evergreen?

**Response:**

Thanks for your valuable comments. P. massoniana, C. lanceolata, and Eucalyptus are evergreen, and HWB, SWB, OCF, and MF are dominated by evergreen species with only a very small proportion of deciduous species in Fujian province. Among all samples used to build the NPP-age relationships, 97.77% were evergreen and only 2.23% deciduous. With such limited samples, reliable growth curves for deciduous types could not be derived. Therefore, we applied a foliage turnover ratio smaller than 1 for all seven forest species.

The manuscript was revised accordingly.

**"2.3.1. Building NPP-age relationships for different forest species**

....

The detailed values for the coefficients of SLA,  $t_l$ , and  $R_{fr,l}$  for different forest species were provided in Table 2 (Li et al., 2024a; Li et al., 2007; White et al., 2000; Xie et al., 2022; Zhou et al., 2008). For HWB, SWB, OCF, and MF,  $t_l$  was assigned evergreen-species values because deciduous samples constitute only 2.23 % of the total samples.

....."

**"4. Discussions**

. . . . . . .

Third, the input coefficients for specific leaf area, foliage turnover ratio, and the ratios of the turnovers of fine roots and leaves to NPP used in calculating forest field NPP for diverse forest species may introduce uncertainties into the forest NPP—age relationships. Currently, these coefficients are primarily sourced from literature (Li et al., 2024a; Li et al., 2007; White et al., 2000; Xie et al., 2022; Zhou et al., 2008), with data originating from subtropical provinces in China such as Guangxi (Xie et al., 2022), Jiangxi (Li et al., 2007), and Guiyang (Zhou et al., 2011), as well as from other regions (White et al., 2000). Data from these regions may differ from those in subtropical China, potentially leading to biases in the calculation of forest field NPP and final built NPP—age curves. Moreover, as deciduous samples constitute only 2.23 % of the total samples, HWB, SWB, OCF and MF were assigned evergreen foliage turnover coefficients. Therefore, future studies should prioritize local field measurements of these key coefficients, particularly for deciduous species, to refine the NPP—age relationships and to quantify the age-dependent carbon sequestration capacity of each species more accurately."

**Comments 1.6:**

The period from 1901 to 1985 was used for the spin-up of the initial model parameters. What do you mean? Do model parameters change with time? It is better to change "the initial model parameters" into "carbon pools". What is the role of the BEPS model? It is not clear how the spin-up was implemented.

**Response:**

Thanks for your valuable comments and suggestions. The period from 1901 to 1985 was used to spin up the soil carbon pools, and it was revised. The 30 m NPP generated by the BEPS model for 2015 (Cao et al., 2025) was used as the input reference NPP.

More description about the model inputs, spin-up, and validations was added in section 2.3.2.

**"2.3.2. Forest carbon modeling using the newly built NPP-age relationships**

[revised manuscript text omitted]

| Input data          |                                                | Unit                                   | Spatial resolution | Temporal resolution | Data source                |
|---------------------|------------------------------------------------|----------------------------------------|--------------------|---------------------|----------------------------|
| Climate data        | Precipitation
Temperature
Vapor pressure | mm
°C
hpa                        | 0.5°               | 1901-2023           | CRU TS
4.08             |
|                     | Cloud amount                                   | %                                      |                    |                     |                            |
| Atmospheric         | CO 2 concentration                  | mol mol -1                  | Site scale         | 1960-2021           | Mauna Loa                  |
| composition
data | Nitrogen deposition                            | 10*gN m -2 yr -1 | 1.27°×2.5°         | 1997-2013           | (Gao et al.,
2020)      |
|                     | Forest cover types                             | /                                      | 30m                | /                   | NFI-II                     |
| Vegetation
data  | LAI                                            | $m^2/m^2$                              | 500m               | 2015                | GLOBMAP
LAI V3          |
|                     | Forest age                                     | year                                   | 30m                | 2015                | NFI-II                     |
|                     | Reference NPP                                  | 10 gC m -2 yr -1 | 30m                | 2015                | BEPS (Cao
et al., 2025) |

|                     | NPP-age                      | /      | /       | / | This study                |
|---------------------|------------------------------|--------|---------|---|---------------------------|
|                     | relationship curves          | /      | /       | / | This sinay                |
|                     | Sand content                 | %      | 0.0083° | / | HDSW                      |
| Soil data           | Clay content                 | %      | 0.0083° | / | World Soil                |
|                     | Soil depth                   | 100 m  | 0.0083° | / | Database                  |
|                     | Latitude/longitude           | degree | 30m     | / | /                         |
| Topographic
data | DEM                          | m      | 30m     | / | http://www.g
scloud.cn |
|                     | Slope and aspect             | /      | 30m     | / |                           |
|                     | Topographic
wetness index | /      | 30m     | / | Calculated
from DEM    |
|                     | Water table depth            | m      | 30m     | / |                           |

**Response to Reviewer #2's comments**

(Italic indicates the manuscript text, red indicates revisions)

**Comments 2.1:**

Accurately establishing the relationship between Net Primary Productivity (NPP) and forest age is a crucial prerequisite for precisely simulating ecosystem carbon sequestration capacity. This study presents the first attempt to establish this relationship at the species scale, with a specific focus on improving NPP prediction accuracy for mature forests. Validation demonstrated that the NPP-age relationship based on the species scale effectively enhanced the accuracy of aboveground biomass (AGB) estimates simulated by the ecological model. This research holds significant scientific merit. The paper addresses an appropriate topic, features a rigorous experimental design, provides thorough argumentation, and maintains a well-structured presentation. It is recommended for acceptance after minor revisions.

**Response:**

Thanks for your positive feedback.

**Comments 2.2:**

Figure 1a: Since the variable represented is categorical, the use of a color gradient is not recommended.

**Response:**

Thanks for your valuable comments. It was revised.

"Figure 1: The distribution of forest species in Fujian Province (a) and the distribution of NFI-I, NFI-II, and SPPCB field survey samples (b). Different colours indicate different forest species, and the grey colour is for bamboo. P. massoniana: Pinus massoniana, C. lanceolata: Cunninghamia lanceolata, Eucalyptus: Eucalyptus robusta smith, HWB: Hardwood Broadleaf excluding Eucalyptus, SWB: Softwood Broadleaf, OCF: Other Coniferous Forests excluding P. massoniana and C. lanceolata, MF: Mixed Forests."

**Comments 2.3:**

AGB Validation Data: The specifics of the AGB data used for validation need further clarification. This includes details on how the data was acquired and its spatial scale. Furthermore, regarding the InTEC model simulation, the grid size employed must be explicitly stated. This is particularly important given the vastly differing spatial resolutions of the model input data (Table 3). Clarify how these data were harmonized to a common scale for simulation. Additionally, address whether scale discrepancies exist between the model-simulated AGB and the ground-collected AGB, and if so, how these were reconciled.

**Response:**

Thanks for your valuable comments. Additional descriptions regarding the AGB validation, the resolution harmonization of model inputs, and the scale discrepancies between the simulated AGB and the ground AGB were added in sections 2.2 and 2.3.2.

A discussion on the varying spatial resolutions of model inputs was also included.

**"2.2. Data**

. . . . . .

[revised manuscript text omitted]

---

## Author Response (AR2)

**Response to Editor's comments:**

(*Italic* indicates the manuscript text, *red* indicates revisions)

**Comments E1:**

Thank you for considering all reviewers' and my suggestions!

I have just a few minor comments on the current version:

- in header of Table 1: If I understand correctly, 1 hm $^2$  = 1 ha. Maybe use here 't ha $^(-1)$ ' as unit.

**Response:**

Thanks for your helpful comments, and it was revised as 't ha-1'.

**Comments E2:**

- Table 3: Please change the unit of LAI to 'm^2 m^(-2)' to be consistent throughout the manuscript.

**Response:**

Thanks for your helpful comments, and it was revised as 'm2 m-2'.

**Comments E3:**

- line 232: Please clarify what you mean with 'peak NPP age'. I guess it should be the age of the forest at its maximum NPP following the derived functions.

**Response:**

Thanks for your helpful comments, and it was revised.

"The age of the forest at its maximum NPP (referred to as the peak NPP age), a critical indicator of the NPP-age relationship, ....."

**Comments E4:**

- lines 263-264: Please clarify, if the 'final forest NPP-age curves' somehow differ from the curves described in section 3.1 and Figure 3.

**Response:**

Thanks for your helpful comments, and it was revised.

"The final species-specific forest NPP-age curves were selected from the built curves using all field NPP samples (green lines in Fig. 3), and their coefficients were provided in Table 4. To facilitate a comparative characterization of forest NPP-age relationships among different forest species, these curves were normalized and jointly displayed in Fig. 5."

**Comments E5:**

- lines 333-334: If you compare the simulated AGBs between the different curves, how do you know that one 'outperforms' the other? How do you know that the newly estimated AGBs are better? Please mention here again your validation data.

**Response:**

Thanks for your helpful comments, and it was revised.

"The built species-specific NPP-age curves were incorporated into the InTEC model for forest biomass modeling. But due to the lack of field soil carbon data for validation, we primarily focused on validating the modeled forest AGB. We compared the simulated AGB obtained by using the newly constructed species-specific NPP-age curves with that obtained by using the previously built nationwide NPP-age curves (Fig.7). Accuracy was evaluated with R² and RMSE against the calculated field AGB from a randomly withheld 20 % of the forest field samples, and higher R² and lower RMSE indicate better performance. Overall, the species-specific NPP-age curves significantly outperformed the nationwide curves in simulating AGB accuracy."